# Complications of Long Head of the Biceps Tenotomy in Association with Arthroscopic Rotator Cuff Repair: Risk Factors and Influence on Outcomes

**DOI:** 10.3390/jcm11195657

**Published:** 2022-09-26

**Authors:** Riccardo Ranieri, Marko Nabergoj, Li Xu, Pierre Le Coz, Ahmad Farihan Mohd Don, Alexandre Lädermann, Philippe Collin

**Affiliations:** 1Department of Biomedical Sciences, Humanitas University, Pieve Emanuele, 20090 Milan, Italy; 2Valdotra Orthopaedic Hospital, 6280 Ankaran, Slovenia; 3Faculty of Medicine, University of Ljubljana, Vrazov Trg 2, 1000 Ljubljana, Slovenia; 4Beijing Jishuitan Hospital, Beijing 100035, China; 5CHU de Rennes, 2 Rue Henri le Guilloux, 35000 Rennes, France; 6UKM Medical Center, Kuala Lumpur 56000, Malaysia; 7Division of Orthopaedics and Trauma Surgery, Geneva University Hospitals, 1205 Geneva, Switzerland; 8Division of Orthopaedics and Trauma Surgery, La Tour Hospital, 1217 Meyrin, Switzerland; 9Faculty of Medicine, University of Geneva, 1205 Geneva, Switzerland; 10Clinique Victor Hugo, 5 Bis Rue du Dôme, 75116 Paris, France

**Keywords:** complication, PROMs, popeye, cramps, shoulder, tendinopathy, arthroscopy

## Abstract

*Background:* This study aims to report the rate of biceps-related complications after LHB tenotomy, investigating related risk factors and their influence on the outcome. The hypothesis is that these complications have a limited clinical influence. *Methods:* A single-center prospective observational study was performed between 2015 and 2017 on consecutive patients who underwent RCR associated with LHB tenotomy. Patients were clinically and radiologically evaluated preoperatively, at six months and one year, and screened for postoperative popeye deformity, cramps, and bicipital discomfort. Each complication was analyzed for the following risk factors: age, sex, body mass index (BMI), dominant arm, manual work, tear patterns, and tendon healing. Finally, the clinical outcome was compared between patients with and without complications. *Results:* 207 patients were analyzed. Cramps, popeye deformity, and discomfort, were respectively, present in 16 (7.7%), 38 (18.4%) and 52 (25.1%) cases at six months and 17 (8.2%), 18 (8.7%) and 24 (11.6%) cases at one year. Cramps were associated with lower age (*p* = 0.0005), higher BMI (*p* = 0.0251), single tendon tear (*p* = 0.0168), manual work (*p* = 0.0086) at six months and manual work (*p* = 0.0345) at one year. Popeye deformity was associated with male sex at six months (*p* < 0.0001). Discomfort was associated with lower age (*p* = 0.0065), manual work (*p* = 0.0099), popeye deformity (*p* = 0.0240) at six months and manual work (*p* = 0.0200), single tendon tear (*p* = 0.0370), popeye deformity (*p* = 0.0033) at one year. Patients without complications showed a significant higher Constant score, pain and subjective shoulder value (SSV) (75.3 vs. 70.4, *p* = 0.00252; 0.9 vs. 1.9, *p* < 0.00001; 80.2 vs. 76.4; *p* = 0.00124) at six months and pain and SSV (0.6 vs. 2.0; *p* = 0.00044; 91.1 vs. 77.8; *p* ≤ 0.00001) at one year. *Conclusions:* Younger age, male sex, higher BMI, manual work, and single tendon tears are risk factors associated with the development of biceps-related symptoms during the first year after tenotomy in association with rotator cuff repair. Nevertheless, the clinical influence of these symptoms on shoulder outcomes is limited.

## 1. Introduction

Disorders of the long head of the biceps (LHB) include instability (subluxation or dislocation), tendinitis, and partial tears, and are commonly associated with rotator cuff tears [1,2,3,4,5]. Due to this strong correlation, in patients with rotator cuff tears, shoulder pain and dysfunction are commonly related to biceps disorders. Walch et al. noticed that the spontaneous rupture of the LHB resulted in pain relief in patients with rotator cuff tear and reported that isolated arthroscopic tenotomy leads to a high degree of patient satisfaction [6]. Consequently, several authors proposed tenotomy or tenodesis as an adjuvant procedure with rotator cuff repair [3,6,7,8].

Many studies aimed to compare tenodesis and tenotomy [8,9,10,11], while only a few studies focused specifically on tenotomy in combination with rotator cuff repair (RCR), trying to consider specific risk factors [12,13,14]. Common drawbacks related to tenotomy include the occurrence of the popeye sign, bicipital cramps, and reduced elbow strength [6,15,16]. In contrast, advantages are related to a quicker recovery, ease of execution, reduced complications, and lesser costs [6,9,12,13]. A meta-analysis of Level I studies showed similar improvements in patient-reported and functional outcomes for these two procedures, with a moderate increased rate of cosmetic deformity in patients undergoing LHB tenotomy [9].

This study aims to highlight the short-term outcome in a large series of patients undergoing LHB tenotomy in arthroscopic RCR, focusing on the occurrence of LHB-related complications. Furthermore, patients with and without complications were compared and analyzed to identify risk factors associated with complication development and to assess differences in clinical outcomes. The hypothesis was that, despite the rate of complication’ occurrence, their clinical influence is limited.

## 2. Methods

Between May 2015 and June 2017, all patients who had a primary RCR performed by the senior author (P.C.) were considered potentially eligible for inclusion in this prospective study. The inclusion criterion was RCR associated with LHB tenotomy based on injury to the pulley and LHB tendinopathy. The exclusion criteria were (1) intra-articular absence of LHB at surgery due to pre-existing tendon tear or congenital variants, (2) LHB adhesions [17], (3) preserved biceps in case of an intact pulley, (4) tenotomy associated with tenodesis, (5) isolated tenotomy in the absence of RCR (because of isolated LHB disease without rotator cuff tear or irreparable massive cuff tear). The study was performed according to the Declaration of Helsinki principles, was approved by the Comité d’Ethique de Recherche Clinique Vivalto Santé (CERC—IRB) (IRB number CERC-VS−2020−09−1), and all patients gave informed written consent.

The outcome of interest was the rate of biceps-related complications (popeye sign, cramps, discomfort) associated with RCR at six months and one year. Furthermore, a comparison of patients with and without LHB-related complications was performed based on various clinical scores. The following baseline characteristics were assessed: age, sex, arm dominance, operated side, body mass index (BMI), and occupation (manual or not manual).

### 2.1. Surgical Technique

A consistent operative technique was used during the study period regarding subscapularis [18], and posterosuperior cuff tear [19]. Repairs were performed with patients in the beach-chair position. Rotator cuff tear was classified as isolated supraspinatus, isolated subscapularis (superior part or total), or using the ABCDE classification described by Collin et al. [20] in case of massive lesions [21]. Intraoperatively, whatever cuff repair status and technical modality of repair, LHB stability was assessed with a probe, its position and macroscopic aspect were precisely analyzed and noted in the surgical report as follows: normal, unstable, inflamed, dislocated, delaminated/torn. LHB tenotomy was performed in the presence of any signs of biceps pathology, including instability, inflammation, or degeneration; in doubtful cases, a tenotomy was performed. Technically, the tenotomy was performed at the glenohumeral joint line, resecting the tendon stump at the supraglenoid tubercle. In the clear absence of pathologies and pulley lesions, the LHB was preserved. No tenodesis associated with tenotomy was performed during the index period. Postoperatively, all patients followed the same standardized rehabilitation protocol [22]. The use of a 20°-abduction sling was discontinued after three weeks. During the first six weeks, patients performed progressive passive overhead stretches and external rotation with the arm at the side. Active range of motion (ROM) started at six weeks, and progressive strengthening at three and four months for non-massive and massive RCR, respectively. Concentric elbow flexion and supination exercises were allowed six weeks postoperatively.

### 2.2. Clinical Evaluation

The postoperative assessment was performed at six months and one year by an independent observer trained in shoulder rehabilitation. At six months, all patients were screened clinically for popeye signs, bicipital cramps, and discomfort referred at the bicipital groove or muscle belly. Assessment parameters comprised pain on the visual analog scale (VAS); active ROM in anterior forward flexion, external rotation with the elbow at the side, and internal rotation behind the back as the nearest spinal level was reached with the thumb; Constant score [23] and its subcategory, the subjective shoulder value (SSV) satisfaction score [24]. Healing was assessed in the repaired rotator cuff tendons on ultrasound, using Sugaya’s classification at six months [25,26]. The tendon was considered healed in types I, II, and III and non-healed in types IV and V. At one year, all patients were interviewed by a phone call to assess pain (VAS obtained by a telematics system), SSV, bicipital cramps, discomfort located on the bicipital groove or muscle belly, and “referred” cosmetic deformity or popeye sign.

### 2.3. Statistical Analysis

For risk factors analysis, patient age and BMI were treated as continuous variables, sex (M/F), operated dominant arm (yes/no), work level (manual/nor manual), tear kind (single isolated tendon tear -supraspinatus or subscapularis-/massive cuff tear -A, B, C, D, E-) and tendon healing (yes/no) were dichotomized. Odds Ratio for Age and BMI expressed the times the odds of related complications for each 1-point increase. For Age and BMI differences between patients with and without the relative complications the as mean ± standard deviation are reported. The association between categorical patient-related factors and postoperative complications was assessed with the chi-square or Fisher’s exact test, as appropriate. If one observed value was ≤5, a Yates’s correction was applied. Crude odds ratios and 95% confidence intervals (CI) were calculated (if one observed value was 0, only frequencies and chi-squared results were reported). A *t*-test, Mann-Whitney U or Wilcoxon Signed-Rank test were used to explore differences for continuous variables, as appropriate. All *p* values were 2-sided with an alpha of 0.05.

## 3. Results

During the study period, 268 patients underwent arthroscopic shoulder surgery for rotator cuff tear and/or biceps disorders: eight patients were excluded because of absent LHB at surgery, 34 patients were excluded because they did not receive any biceps procedures, 17 patients were excluded due to an isolated biceps procedure without cuff repair, and two patients were lost at final follow-up, resulting in 207 patients undergoing tenotomy in association with rotator cuff repair, available for analysis (Figure 1).

Demographic data and clinical results are reported in Table 1.

Types of rotator cuff ruptures and relative frequencies were reported in Figure 2.

The status of LHB and relative frequencies were reported in Figure 3. 

Considering all patients, active anterior forward flexion increased from mean 132.4° ± 33.9° (range, 10° to 180°) to 153.1° ± 20.6° (range, 80° to 100°) (*p* < 0.00001), external rotation from 42.1° ± 19.3° (range 0° to 90°) to 49.3° ± 17.7° (range 0° to 90°) (*p* < 0.00001), internal rotation from a median of L5/L3 (range, buttock to T12) to T12 (range, buttock to T12) (*p* = 0.0124); Constant score from 54.3 ± 16.4 (range, 20 to 93) to 73.6 ± 12.5 (range, 28 to 100) (*p* < 0.00001); pain (VAS) from 4.6 ± 1.9 (range, 0 to 9.7) to 1.3 ± 1.5 (range, 0 to 8) (*p* < 0.00001) to 0.9 ± 1.4 (range, 0 to 6.3) (*p* = 0.0001); SSV from 50.8 ± 17.6 (range, 5 to 90) to 77.4 ± 14.7 (range, to 30 to 100) at six months (*p* < 0.00001) and to 88.5 ±13.1 at one year (range, 45 to 100) (*p* < 0.00001). In 165 (79.7%) patients, repaired tendons were considered healed at six months; among 149 patients with single tendon tears, 121 repairs healed (81.2%), and among 58 patients with massive cuff tears, 44 (75.8%) repairs healed.

Patients presenting LHB-related complications and relative rates of each complication at six months and one year, with relative *p*-value, were reported in Table 2. 

No patients required revision surgery to convert the LHB tenotomy to a tenodesis for postoperative cosmetic deformity. Exploration of risk factors’ association with each complication at six months and one year is presented in Table 3, reporting the relative odds ratio with the 95% CI and the *p*-value.

At six months, every one-year increase in age was associated with 0.88 (95% CI, 0.81–0.96) and 0.94 (95% CI, 0.90–0.98) times the odds of referring cramps and discomfort, respectively, and every 1-point increase in BMI with 1.1 (95% CI, 1.02–1.23) times the odds of referring cramps. Patients with cosmetic deformity had 2.3 (95% CI, 1.1–4.9) and 3.4 (95% CI, 1.1–10.7) times more frequent reporting of biceps discomfort at six months and one-year follow-up, respectively. Male patients had 6.9 (95% CI, 3.0–15.5) times the odds of objective cosmetic (popeye) deformity at six months. Patients who had a manual work had 5.0 (95% CI, 1.5–16.0) and 2.9 (95% CI, 1.1–8.3) times the odds of referring cramps at six months and one-year follow-up, respectively, and had 2.3 (95% CI, 1.2–4.3) and 2.8 (95% CI, 1.1–6.6) times the odds of reporting biceps discomfort at six months and one-year follow-up, respectively. Patients with a single tendon tear had 4.9 (95% CI, 1.1–21.3) times the odds of reporting biceps discomfort at one year and had a higher incidence of cramps at six months (16/133 vs. 0/58, *p* = 0.0168). No other significant risk factors related to complications were found.

Patients who presented at least one complication did not show significant differences (*p* > 0.05) regarding preoperative functional outcomes, distribution of tear types, or healing rates compared to patients who did not present any complications at six months or one year. Data regarding the comparison of patients who developed at least one complication with patients who did not experience any complications at six months and one year are presented in Table 4.

## 4. Discussion

Two main findings emerged from this paper. First, the complication rate following tenotomy was not irrelevant and increased in cases of male sex, younger age, higher BMI, manual work and isolated single tendon tear. Second, patients who presented at least one biceps complication showed statistically significant lower clinical outcomes at six months and one year, but this difference has a limited clinical influence, confirming our hypothesis. These results may help surgeons in the decision to perform an LHB tenotomy in association with an RCR.

LHB tenotomy is a relatively simple and inexpensive procedure to manage biceps disorders associated with rotator cuff disorders, providing good results [3,4,6,7,8,17]. However, biceps-related complications may occur [6,12,13,14]. In this series, 75 (36%) and 41 (20%) patients reported at least one complication at six months and one year, respectively, with an overall significant decrease of frequency between the two follow-ups. Moreover, the rate of single complications decreases significantly from six months to one year, except for bicipital cramps. This is the first study to follow the evolution of biceps-related complications at two consecutive follow-ups: these findings could be beneficial for clinicians who face patients that developed these complications immediately after surgery, explaining to the patient that a spontaneous symptom improvement could be expected at one year.

Popeye deformity was reported in 38 (18.4%) cases at six months, and 18 (8.7%) cases at one year. It is essential to notice that the modality of evaluation between the two follow-ups was different. At six months, a shoulder-trained physiotherapist objectively evaluates the deformity, while at one year, the patients were asked to refer any subjective deformity of the arm. At least two factors are responsible for these findings. First, the popeye deformity is often a subtle sign unnoticed by patients. The incidence at six months is similar to other studies [12,13], while the “subjective” reported deformity rate seems to be lower. This could mean that the importance of the detection of the popeye sign by the clinician is overestimated compared to the patient’s perception. This concept of little patient concern for deformity was also underlined by other authors [6,12], in particular by van Deurzen et al. [27], and is reinforced by the findings of this study. Second, patients were allowed to perform muscular reinforcement between three and six months. With time, such strengthening may partially restore the shape of the arm. However, in this series, it is also true that at one year, the presence of a referred deformity is associated with a higher risk of referred bicipital discomfort, remembering not to underestimate that the occurrence of this complication may influence the patient satisfaction at one year.

Male sex was found to be associated with developing a cosmetic deformity, a finding also reported by a previous study [13,14,27,28]. Even though some authors suggested tenodesis in thin patients due to the high probability of cosmetic deformity and concerns [29], in this series the development of an “objective” or “subjective” popeye deformity was not associated with BMI. Similar findings regarding BMI were also reported by Lim et al. [14] and Mirzayan et al. [13]. On the contrary, in this study, higher BMI was associated only with cramping at six months after tenotomy. Despite these findings, the common practice to avoid simple tenodesis in young and thin subjects should still be taken into consideration, because these patients may be more bothered by a cosmetic deformity for an esthetical reason, even if the occurrence is low.

The development of biceps discomfort and cramps was found to be significantly associated with lower age, manual work, and single tendon tear. Similar findings regarding age were also found by Kelly et al. [28] and Mirzayan et al. [13], even if, in this series, younger age emerged as a significant risk factor associated with these symptoms at six months but not at one year.

In the present series, tenotomy in combination with the repair of an isolated supraspinatus or subscapularis tear was associated with a higher risk of cramps at six months and discomfort at one year. Some authors suggested an auto-tenodesis effect in the bicipital groove as a possible explanation of the lack of biceps symptoms after a tenotomy [13,30]. Additionally, Ahmad et al. showed in their cadaveric study that a diseased LHB tendon, with increased flattening and broader dimensions, resulted in greater forces to drop through the bicipital groove [31]. As a more pathological bicipital tendon is associated with an increase of rotator cuff tear size [2,4], it is possible that, in a massive cuff tear, the auto-tenodesis phenomenon is more likely to occur due to an increased LHB hypertrophy. By this theory, a similar finding was also reported by Mirzayan et al. [13], who showed a positive trend in the association between hypertrophy and no cramping. As pinpointed by Godeneche et al., adjuvant LHB tenodesis might consequently be required when repairing isolated supraspinatus tears [8].

The findings regarding manual work are very interesting. Several authors recommended tenodesis over tenotomy for more active patients [4,10,14,29,32]. Still, the higher risk of biceps-related complications in manually active patients was never reported, probably because, excluding sports patients or athletes, it is difficult to quantify the activity level of a person. Categorizing the kind of occupation in manual and not manual work was a significant risk factor for the development of biceps cramps and discomfort. A previous study about biceps-related complications after tenotomy investigated a similar factor, sub-analyzing patients in the “manually active” group and a “sedentary” group, but no significant differences were found [12]. We do not have a clear explanation about our different results: it is possible that the distinction between manual and not manual work was more accurate in this series, or that with longer follow-up (mean 40 months in the study by Duff and Campbell [12]) the incidence of biceps related symptoms, such as cramps and discomfort, may change over time and decrease in manual patients. Regardless, based on the findings of this study, patients who performed a manual occupation should be warned of the higher probability of experiencing bicipital cramps or discomfort during the first year following a biceps tenotomy.

The most important finding of this study emerged from comparing clinical outcomes between patients who experienced at least one biceps-related complication and patients without any complications. Despite a statistically significant difference of almost all considered parameters, the difference seems not clinically relevant (under the minimal clinically important difference for CS and VAS in the setting of rotator cuff surgery [33,34]). The persistence of biceps-related complications at one year may negatively influence the subjective perception of the shoulder.

The strengths of this study include the prospective observational design, the largest series to date, its monocentric nature (each patient was treated by the same surgeon using the same validated technique and following the same rehabilitation program), the systematic ultrasound assessment of rotator cuff healing at six months, and the fact that both evaluations were performed by an independent and trained physiotherapist. However, this study presents some limitations. First, a tenodesis control group was not included because no tenodesis was performed during the index period, and the aim of the study was to analyze the complications rate and the clinical impact of tenotomy associated with RCR. Second, the analysis was performed at a short-term follow-up (six months and one year). The decision to conduct a short-term study was because the most biceps-related complications usually occur during the first year after surgery [35]. In this one-year follow-up prospective study, the number of lost patients is expected to be very limited (two patients in this series). Moreover, LHB tenotomy does not alter the natural history of shoulder disease at longer follow-up [6]. However, it will be interesting to follow the patients in mid-and long-term follow-up. Third, an objective evaluation of elbow flexion or supination strength whose decrease has been associated with LHB rupture or tenotomy was not performed. Based on previous studies focusing specifically on tenotomy, the objective amount of strength reduction is minimal [9,11,12], and based on our clinical experience, excluding athletes or sports patients, it is not clinically relevant. The fourth limitation, the LHB morphology at arthroscopic evaluation, was not considered in the risk factor analysis, which is different from previous studies [13], because we feel that this classification is too subjective and was never validated. Finally, to have a sufficient number of patients in both groups, the comparison of clinical outcomes was performed considering the presence or not of any complications rather than each complication singularly.

## 5. Conclusions

Younger age, male sex, higher BMI, manual work, and single tendon tears are risk factors associated with the development of biceps-related symptoms during the first year after tenotomy in association with rotator cuff repair. Nevertheless, the clinical influence of these symptoms on shoulder outcomes is limited. When discussing with a patient about biceps tenotomy in rotator cuff surgery, beyond considering these risk factors to propose a tenodesis, it should be explained that the shoulder function is minimally influenced in case of popeye deformity, bicipital cramps, and/or discomfort at short-term follow-up.

## Figures and Tables

**Figure 1 jcm-11-05657-f001:**
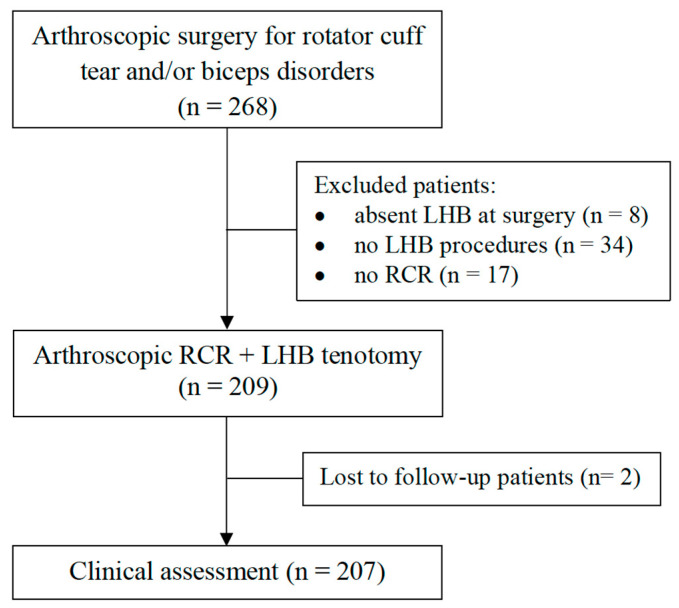
Flowchart. LHB, long head of the biceps; RCR, rotator cuff repair.

**Figure 2 jcm-11-05657-f002:**
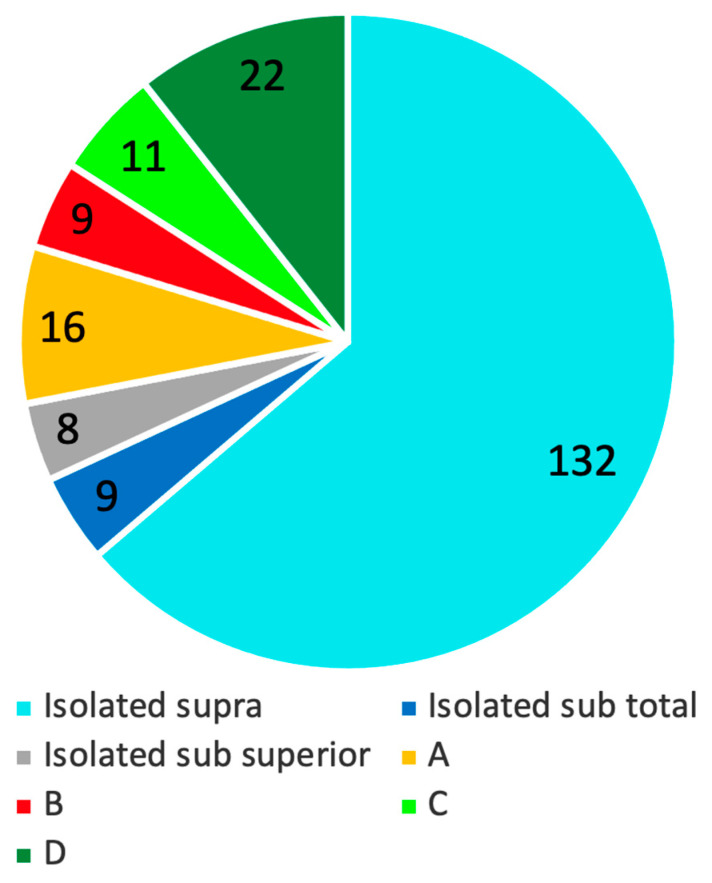
Tears distribution. A, supraspinatus and superior subscapularis tears; B, supraspinatus and entire subscapularis tears; C, supraspinatus, superior subscapularis, and infraspinatus tears; D, supraspinatus and infraspinatus tears.

**Figure 3 jcm-11-05657-f003:**
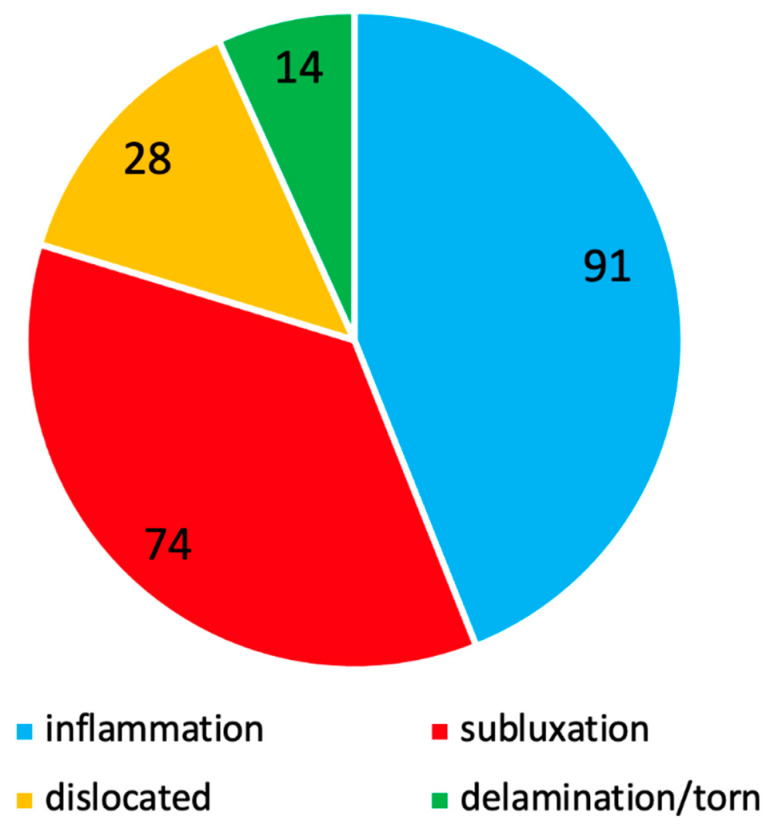
Biceps pathologies distribution.

**Table 1 jcm-11-05657-t001:** Demographic data.

Variable	Mean ± SD or Number of Observations (%)
Age (years)	60.6 ± 8.1
Male/Female	83 (40.1)/124 (59.9)
BMI	27.4 ± 4.7
Dominant shoulder operated	151 (73.7)/56 (27.3)
Manual work	84 (40.6)/123 (59.4)

BMI, body mass index.

**Table 2 jcm-11-05657-t002:** LHB-related complications.

LHB Related Complications	6 Months N. Observation (%)	1 Year N. Observation (%)	*p* Value
At least one complication	75 (36.2)	41 (19.8)	<0.001 *
Cramps	16 (7.7)	17 (8.2)	0.856
Cosmetic deformity ^	38 (18.4)	18 (8.7)	0.004 *
Biceps discomfort	52 (25.1)	24 (11.6)	<0.001 *

LHB, long head of the biceps; ^ “objective” evaluation at 6 months and “subjective” referred at 1 year; * *p* < 0.05.

**Table 3 jcm-11-05657-t003:** Risk factors for complications.

	Cramps			Popeye			Discomfort		
Risk Factors	OR	95% CI	*p* Value ^	OR	95% CI	*p* Value ^	OR	95% CI	*p* Value ^
Age 6 months 1 year	0.88	0.81–0.96	<0.001 *	0.95	0.95–1.03	0.660	0.94	0.90–0.98	0.007 *
	54.5 ± 5.9 vs. 61.1 ± 8.0			60.3 ± 8.3 vs. 60.6 ± 8.0			57.8 ± 7.3 vs. 61.5 ± 8.1		
	0.98	0.92–1.05	0.412	1.02	0.96–1.1	0.610	0.96	0.90–1.01	0.159
	59.4 ± 8.2 vs. 60.7 ± 8.1			61.7 ± 8.5 vs. 60.4 ± 8			58.2 ± 7.5 vs. 60.9 ± 8.1		
BMI 6 months 1 year	1.1	1.02–1.23	0.025 *	0.98	0.9–1.1	0.912	1.1	0.98–1.1	0.190
	30 ± 5.2 vs. 27.2 ± 4.7			27 ± 3.3 vs. 27.5 ± 4.5			28.2 ± 5.2 vs. 27.1 ± 4.6		
	0.97	0.9–1.1	0.529	1.00	0.9–1.1	0.849	0.99	0.9–1.1	0.920
	26.7 ± 4.7 vs. 27.5 ± 4.8			27.5 ± 4.2 vs. 27.4 ± 4.8			27.2 ± 4.1 vs. 27.4 ± 4.8		
Male sex									
6 months	0.9	0.3–2.5	0.825	6.9	3.0–15.5	<0.001 *	0.9	0.5–1.7	0.781
1 year	0.6	0.2–1.8	0.491	1.6	0.6–4.1	0.370	0.7	0.3–1.8	0.472
DA operated									
6 months	0.5	0.2–1.3	0.145	0.8	0.4–1.7	0.589	0.9	0.5–1.9	0.878
1 year	1.2	0.4–4.0	0.714	0.7	0.3–2.0	0.549	2.0	0.7–6.1	0.305
Manual work									
6 months	5.0	1.5–16.0	0.009 *	1.4	0.7–2.9	0.346	2.3	1.2–4.3	0.010 *
1 year	2.9	1.1–8.3	0.034 *	1.9	0.7–5.1	0.176	2.8	1.1–6.6	0.020 *
STT vs MCT									
6 months	-°	-°	0.0168 *	0.8	0.4–1.7	0.589	1.9	0.9–4.1	0.103
1 year	3.1	0.7–14.1	0.120	1.0	0.3–3.0	0.981	4.9	1.1–21.3	0.037 *
Healed tendon									
6 months	0.5	0.1–2.5	0.608	1.5	0.7–3.5	0.307	0.5	0.2–1.3	0.157
1 year	1.2	0.4–4.0	0.485	0.2	0.03–1.6	0.165	0.8	0.2–2.4	0.836
Popeye									
6 months	2.2	0.7–6.7	0.065	-	-	-	2.3	1.1–4.9	0.024 *
1 year	2.5	0.6–9.7	0.173	-	-	-	3.4	1.1–10.7	0.003 *

DA, dominant arm; BMI, body mass index; STT, single tendon tear; MCT, massive cuff tear; ^ Chi-squared or Mann-Whitney U; * *p* < 0.05; -° OR not calculated because one observed value was 0.

**Table 4 jcm-11-05657-t004:** Comparison of patients with and without complications.

	No LHB Complications	≥1 LHB Complications	*p* Value
	six months		
N ^ cases	132	75	
Isolated tendon/MCT	92 (69.7%)/40 (30.3%)	57 (76%)/18 (24%)	0.332
Healed/non-healed	105 (79.5%)/27 (20.5%)	60 (80%)/15 (20%)	0.938
SS involvement ^1^	101 (76.5%)	53 (70.7%)	0.447
AE postop	154.2 ± 23.7	149.1 ± 22.0	0.039 *
ER postop	50.3 ± 18.4	47.5 ± 16.4	0.250
IR postop (median)	T12	L5/L3	0.017 *
Constant score postop	75.3 ± 11.3	70.4 ± 13.9	0.003 *
Pain (VAS)	0.9 ± 1.3	1.9 ± 1.6	<0.001 *
SSV	80.2 ± 13.2	76.4 ± 17.0	0.001 *
	1 year		
N ^ cases	166	41	
Isolated tendon/MCT	116 (69.9%)/50 (30.1%)	33 (80.5%)/8 (19.5%)	0.244
Healed/non-healed	130 (78.3%)/36 (21.7%)	35 (85.4%)/6 (14.6%)	0.390
Pain (VAS)	0.6 ± 1.0	2.0 ± 2.0	<0.001 *
SSV	91.1 ± 11.2	77.8 ± 15	<0.001 *

N ^, number of; LHB, long head of the biceps; MCT, massive cuff tear; AE, anterior elevation; ER, external rotation; IR, internal rotation; VAS; visual analogue scale; SSV; subjective shoulder value; * *p* < 0.05; SS, scubscapularis; ^1^ SS involement = isolated subscap, type A, B or C tears.

## Data Availability

Not applicable.

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
