# Peer review of "Complications of Long Head of the Biceps Tenotomy in Association with Arthroscopic Rotator Cuff Repair: Risk Factors and Influence on Outcomes"

_jcm, 2022, doi:10.3390/jcm11195657_

Round 1
Reviewer 1 Report
Dear editoral board of JCM,
Thank you kindly for the opportunity to review this interesting study about biceps tenotomy during rotator cuff repairs.
I would recommend a few changes to strengthen the paper.
line 46,47. not sure if you can make this statement: shoulder pain is related to biceps disorders.
line 47-50. This is an important study but because spontaneous biceps rupture reliefs pain we should operate on everyone. I am missing some background on different hypotheses and functions of the LHBT. the hourglass vs depressor function. or from an anatomical point of view: intra-articular course, remaining structure in case of cuff tear. For example this study: doi 10.1097/CORR.0000000000001555 investigates the role of the biceps more profoundly.
line 51: double reference 7
line 87: ....posterosuperior cuff tears.
105: any follow-up more than 1 yr?
figure one: it's not that patients are lost, but lost to follow-up.
line 166: table 2 should be placed after line 177. it's now within a sentence.
line 202: in general the discussion is quite long. it should be condensed.
line 207: it is concluded that the difference is VERY limited. but the differences are statistically significant. Although explained further on with MCID, it may need some explanation at this place as well.
line 243-246: in current times it is not 'woke' to only judge females. rather leave out.
254: would only work in small tears?
275: a longer FU would make a good explanation.
297: by an independent and trained physiotherapist.
conclusion: line 321, suggestion to leave out 'very'. because it is still a difference.
another ref of great value: https://doi.org/10.1016/j.jse.2020.10.040 please reference and make a comment in the discussion section. How much is the popey sign relevant?
With kind regards.
Reviewer 2 Report
Thank you for the opportunity to review this study for Journal of Clinical Medicine.
In this study, the authors reported the risk factors of biceps-related complications after rotator cuff repair with LHB tenotomy and the effect of biceps-related complications on shoulder functional outcomes. This study has the great advantage of including a large sample size among studies evaluating biceps-related complications after rotator cuff repair with LHB tenotomy and detailed evaluation of functional outcomes. However, I have several questions and comments about this study, as follows.
Major Comments
1. In this study, the postoperative range of motion of internal rotation in patients with biceps-related complications was 3-5 vertebral levels lower than in patients without this complication, which we consider clinically significant. Therefore, it would be advisable to evaluate not only the presence of massive rotator cuff tear but also the presence of subscapularis tendon tear as a risk factor for biceps-related complication.
2. As this study was conducted using only univariate analysis, a possible limitation is that it may not have eliminated confounding relationships among explanatory variables. For example, although "younger age" and "manual worker" were identified as risk factors for biceps cramp, it is unclear which factor is more influential, since it is thought that a larger proportion of younger people are in manual work occupations. To overcome this limitation, it would be better to consider conducting a multivariate analysis such as logistic regression analysis.
Minor Comments
1. Please unify the significant figures of P-values shown in this study, as they are not unified throughout the manuscript.
2. Please include the missing odds ratio in Table 3, which shows the relationship between 6-month postoperative crumps and tear type.
3. There are two 3) in Method's exclusion criteria, please modify them.
Round 2
Reviewer 2 Report
Thank you for the opportunity to review this study for Scientific Reports. The authors provide sufficient answers and the changes strengthen the manuscript.